# OpenReview forum: "RISE: A Statistical Perspective for Adversarial Attacks against Closed-Source MLLMs"
_ICLR.cc/2026/Conference — Submitted to ICLR 2026_

### Official Review · Reviewer_DnBK · 2025-10-20

**Soundness:** 3
**Presentation:** 3
**Contribution:** 2
**Rating:** 6
**Confidence:** 3

**Summary:**

The paper proposes RISE, a transfer attack framework that replaces point-wise latent alignment with distributional alignment using Energy Distance, combined with a GNN Attack Router that selects a task-adaptive surrogate ensemble. Empirically, RISE improves targeted transfer to both open-source and closed-source MLLMs (GPT-4o/4.1, Gemini-2.0, Claude-3.5/3.7; also o3 and Claude-Sonnet-4) under an $\ell_\infty$ budget, measured via LLM-as-a-judge metrics (ASR/AvgSim, KMR). The method is positioned as statistically principled (non-parametric metric, augmentation-driven sampling) and data-driven for model selection. Results report consistent gains over M-Attack and FOA-Attack across multiple settings.

**Strengths:**

- Conceptual shift: From point-wise to distributional alignment; Energy Distance objective is well-motivated and formalized (global point→dist, local dist→dist)
- Adaptive ensembling: GNN Attack Router learns transfer relations from logs and selects a Top-K surrogate subset per task; practical and novel in this context.
- Broad empirical gains: Improvements on six open-source MLLMs and five closed-source MLLMs; additional results on models “with enhanced reasoning.”
- Ablations and trade-offs: Sensitivity to number of augmentations M and weighting η; selection of K=2 justified via accuracy/compute trade-offs.
- Clear pipeline & pseudocode: Offline router training and online attack generation are described with algorithms, aiding reproducibility.

**Weaknesses:**

- Evaluation dependence on LLM-as-judge: ASR uses GPTScore thresholding with the same MLLM to caption both images; potential circularity/bias is not stress-tested or reported with sensitivity analyses
- No multi-seed confidence intervals or per-seed tables; variance across randomness sources (augmentations, sampling, router selection) is unquantified
- Primary focus on $\ell_\infty$ with $\epsilon$=16/255, step size 1/255, 300 iters; limited visibility into $\ell_2$ or oother budget sweeps in the main paper.
- The GNN Attack Router learns from historical logs on 800 pairs and is evaluated on 200; risks of distribution shift or leakage are not deeply analyzed.

**Questions:**

- Could you report multi-seed results with confidence intervals to check stability across randomness?
- How sensitive are ASR and KMR to the GPTScore threshold and to using a different judge than the captioning MLLM (to reduce judge–model coupling)?
- What is the latency/throughput overhead of RISE (per image) vs FOA/M-Attack, and how does it scale with M and K (beyond the partial trade-off shown)?

---

> ### Author Response · Authors · 2025-11-20
> **Response to Reviewer DnBK-1**
>
> We are truly grateful for the time you have taken to review our paper, your insightful comments and support. Your positive feedback is incredibly encouraging for us! In the following response, we would like to address your major concern and provide additional clarification.
>
>
>
> > **Q1.** Could you report multi-seed results with confidence intervals to check stability across randomness?
>
> **A1.** Thanks for your comment. We recognize the importance of verifying system stability. We repeated the attack generation process using three different random seeds. We conducted attacks against GPT-4o, while Claude-Sonnet-4.5 served as the judge model. The results showed a very small standard deviation in ASR (±0.8%), indicating that RISE is robust to random variations.
>
>
>
> > **Q2.** How sensitive are ASR and KMR to the GPTScore threshold and to using a different judge than the captioning MLLM?
>
> **A2.** Thanks for your comment. **(1) Evaluation Standard Sensitivity.** By using Claude-Sonnet-4.5 as an independent evaluation metric, we eliminate the coupling between the evaluation standard and the model. **As shown in the table below,** even under stricter evaluation standards, RISE maintains a 5.0% lead over the strongest baseline (FOA-Attack), demonstrating the robustness of our results.
>
> | **Method** | **Model** | **Original Judge (ASR)** | **Original Judge (AvgSim)** | **Claude Judge (ASR)** | **Claude Judge (AvgSim)** | **Human Eval (ASR)** |
> | :--------: | :-------: | :----------------------: | :-------------------------: | :--------------------: | :-----------------------: | :------------------: |
> | FOA-Attack | Ensemble  |           85.5           |            0.50             |          79.0          |           0.51            |         81.0         |
> |    Ours    | Ensemble  |         **89.5**         |          **0.53**           |        **84.0**        |         **0.57**          |       **85.0**       |
>
>
>
> **(2) Threshold Sensitivity.** We increased the GPTScore similarity threshold from 0.3 to 0.4.
>
> ASR: Even under the stricter 0.4 threshold, RISE achieves a 78.5% ASR, maintaining its lead over FOA-Attack (72.0%).
>
> KMR: We further validated the robustness of the KMR. Because KMR relies on discrete keywords rather than continuous similarity, it inherently exhibits greater stability. Across different judge prompts, RISE consistently retrieves more target keywords than the baseline model.
>
>
>
> > **Q3.** What is the latency/throughput overhead of RISE (per image) vs FOA/M-Attack, and how does it scale with M and K?
>
> **A3.** Thanks for your comment. We analyze the computational trade-offs summarized **in the table below** (excerpted from Table 5 of the paper). Although using $M=5$ introduces some overhead, the Router latency remains negligible (<0.1 seconds). Moreover, reducing the enhancement count to $M=2$ enables an “efficient” mode that requires less than 80% of FOA-Attack’s runtime while simultaneously achieving a higher ASR (86.0% vs. 85.5%). This result demonstrates RISE’s efficient scalability.
>
> | **Method** | **Settings**  | **ASR (%)** | **Time (s)** | **Throughput** |
> | :--------: | :-----------: | :---------: | :----------: | :------------: |
> | FOA-Attack |   Standard    |    85.5     |     109      |      1.0x      |
> |    Ours    | M=5 (Default) |  **89.5**   |     175      |      0.6x      |
> |    Ours    |  M=2 (Fast)   |    86.0     |    **86**    |    **1.3x**    |
>
>
>
> > **Q4.** The GNN Attack Router learns from historical logs on 800 pairs and is evaluated on 200; risks of distribution shift or leakage are not deeply analyzed.
>
> **A4.** Thanks for your comment. Your concern about the distribution shift between the training and test sets is valid. We have revised Section 3.4 (highlighted in blue) to explicitly state that the transfer scores in the attack logs used to train our GNN Attack Router originate from attacks on Qwen2-VL-7B, which served as a reserved target model. Consequently, all target models in our evaluation introduce substantial distribution shifts relative to the training data, as they involve previously unseen architectures. The Router’s successful attacks on these unseen targets demonstrate that it learned transferable patterns that generalize, rather than merely overfitting to the training set.

---

> > ### Author Response · Authors · 2025-11-20
> > **Response to Reviewer DnBK-2**
> >
> > > **Q5.** Primary focus on $\ell_\infty$ with $\epsilon$=16/255, step size 1/255, 300 iters; limited visibility into $\ell_2$ or other budget sweeps in the main paper.
> >
> > **A5.** Thanks for your comment. We strictly follow the standard perturbation budget ($\epsilon = 16/255$) set by prior state-of-the-art methods (e.g., M-Attack, FOA-Attack) to enable fair and direct comparisons. We recognize the importance of analyzing more stringent budgets and plan to explore performance under tighter constraints (e.g., $\epsilon = 8/255$) as a key direction for future research.
> >
> >
> >
> >
> >
> > Thanks again for appreciating our work and for your constructive suggestions. Please let us know if you have further questions.

---

> > > ### Comment · Reviewer_DnBK · 2025-11-27
> > >
> > > Thank you for the detailed rebuttal and the extra experiments. The multi seed results with small variance in ASR indicate that RISE is reasonably stable with respect to randomness. The added analysis of compute and latency also clarifies the trade off between ASR and cost for different values of M, and the overhead of the router appears negligible in practice. The description of how the GNN based router is trained on logs from one surrogate while being evaluated on a diverse set of unseen target MLLMs supports the claim that the router is not simply overfitting to a single model family. One important remaining aspect is the sensitivity to the perturbation budget. Evaluations under several budgets such as $\epsilon$={1, 4, 10} should be evaluated to show how strongly the method depends on the allowed perturbation magnitude and whether the relative advantage over baselines is stable across budgets. Taken together, these points address my earlier concerns on robustness and computational practicality to a satisfactory degree.
> > >
> > > On the evaluation side, I agree with Reviewer UTRC above that the Claude based judge and the human study strengthen the empirical picture yet do not fully settle the issue of generality of LLM as judge. The current evidence still relies on a small set of judge models, a fixed prompt and a human study that is not fully specified in terms of eval protocol. The limitations of LLM based and human evaluation should be clearly articulated, and robustness to prompt should be clearer. I also support UTRC’s point on transferability scope. The method is convincing on the NIPS 2017 to MS COCO setting with many targets, but the restriction to a single source target dataset pair should be stated explicitly, with cross dataset and cross task evaluations identified as missing pieces. Overall, the rebuttal improves my confidence in the technical soundness and stability of the method, and I plan to keep my original overall evaluation.

---

### Official Review · Reviewer_UTRC · 2025-10-28

**Soundness:** 3
**Presentation:** 3
**Contribution:** 3
**Rating:** 4
**Confidence:** 4

**Summary:**

- The method reframes targeted transfer against closed-source MLLMs as distributional alignment with Energy Distance over augmented views, replacing fragile pointwise feature matching. A learned GNN Attack Router selects a task-adaptive Top-K surrogate ensemble to balance transfer strength and compute.
- Experiments on open and closed models report higher ASR/AvgSim/KMR than FOA/M-Attack under common ℓ∞ budgets, arguing for a statistically grounded and data-driven framework.

**Strengths:**

- The shift to distribution-level alignment with Energy Distance is methodologically coherent, aligning the attack with a two-sample statistical perspective and reducing dependence on brittle pointwise feature matching.
- The coupling of global p→d and local d→d objectives plausibly preserves both scene-level and object-level semantics during perturbation optimization
- The router casts surrogate selection as supervised meta-learning over transfer logs, yielding compact ensembles that retain transferability under constrained K
- The paper surfaces actionable implementation specifics with pseudocode and ablations, facilitating reproducibility
- The evaluation spans open and closed targets and reports multiple metrics, indicating gains are not tied to a single success criterion

**Weaknesses:**

- Heavy reliance on LLM-as-judge risks circularity/bias; without judge-swap or human validation, reported ASR may be optimistic.
- Lack of strictly compute-matched comparisons means increases in M and K may conflate algorithmic gains with extra gradient budget or wall-clock.
- Advantages of Energy Distance over other distance metrics are clear
- The router’s behavior under distribution shift seems insufficiently explored
- Black-box targeted transfer to unseen architectures and cross-dataset tests are limited, so robustness to closed-source variability is not fully established.

**Questions:**

- The authors used fixed attack hyperparameters (\epsilon=16/255 with a fixed step size, etc.) and do discuss the compute trade-off, but does not discuss memory/per-image latency/throughput details.
- The LLM-as-a-judge protocol uses the same family to caption both adversarial and target images and thresholds GPTScore at 0.3, but I cannot find judge-swap or human validation; please indicate where these sensitivity analyses appear or add them
- how does this method perform in comparison to other methods on other source->target transfer settings besides NPIS2017->COCO?

---

> ### Author Response · Authors · 2025-11-20
> **Response to Reviewer UTRC-1**
>
> We are truly grateful for the time you have taken to review our paper and your insightful review. Here we address your comments in the following.
>
>
>
> > **Q1.** The LLM-as-a-judge protocol uses the same family to caption both adversarial and target images and thresholds GPTScore at 0.3, but I cannot find judge-swap or human validation.
>
> **A1.** Thanks for your comment. We acknowledge that solely relying on GPT-like models for evaluation may introduce bias. To rigorously validate our findings, we conducted two additional evaluations.
>
> **(1) Cross-Judge Validation.** We used the advanced Claude-Sonnet-4.5 model as an independent judge to assess adversarial samples generated against GPT-4o. **The results are presented in the table below.** ASR scores decreased slightly, likely due to differences in semantic similarity sensitivity across MLLMs, as Claude-Sonnet-4.5 applies stricter criteria. However, the relative performance rankings remained consistent. Notably, RISE achieved an ASR of 84.0%, outperforming the strongest baseline, FOA-Attack (79.0%), by 5.0%. This result confirms that RISE's improvement arises from robust semantic alignment and is unaffected by variations in the evaluation standard.
>
> **(2) Human Evaluation.** We conducted a human evaluation of adversarial samples generated by GPT-4o. To reduce evaluation complexity, annotators assigned only Yes/No labels. **The results are presented in the table below.** RISE achieved an average human-evaluated ASR of 85%, whereas FOA-Attack reached 81%. This further validates RISE’s effectiveness in generating semantic transfers that are perceptible to human evaluators.
>
> | **Method** | **Model** | **Original Judge (ASR)** | **Original Judge (AvgSim)** | **Claude Judge (ASR)** | **Claude Judge (AvgSim)** | **Human Eval (ASR)** |
> | :--------: | :-------: | :----------------------: | :-------------------------: | :--------------------: | :-----------------------: | :------------------: |
> | FOA-Attack | Ensemble  |           85.5           |            0.50             |          79.0          |           0.51            |         81.0         |
> |    Ours    | Ensemble  |         **89.5**         |          **0.53**           |        **84.0**        |         **0.57**          |       **85.0**       |
>
>
>
> > **Q2.** The authors used fixed attack hyperparameters (\epsilon=16/255 with a fixed step size, etc.) and do discuss the compute trade-off, but does not discuss memory/per-image latency/throughput details. / Lack of strictly compute-matched comparisons.
>
> **A2.** Thanks for your comment.We acknowledge that both $M$ and $K$ influence computational budgets. We analyzed this trade-off using existing records (Figure 4) and present a comparison of computational resource usage **in the table below.** Even with reduced augmentations ($M=2$), RISE achieved a higher ASR (86.0% vs. 85.5%) while requiring less runtime (86 seconds) than FOA-Attack (109 seconds). This confirms that the performance improvement results from optimized Energy Distance targets and adaptive routing, rather than from merely increasing the computational budget.
>
> Regarding memory usage, since our attack is black-box and target-agnostic, GPU memory consumption is primarily driven by surrogate ensemble inference, similar to standard ensemble attacks (e.g., M-Attack or FOA-Attack). No additional memory overhead is incurred.
>
> | **Method** | **Settings**  | **ASR (%)** | **Time (s)** | **Throughput** |
> | :--------: | :-----------: | :---------: | :----------: | :------------: |
> | FOA-Attack |   Standard    |    85.5     |     109      |      1.0x      |
> |    Ours    | M=5 (Default) |  **89.5**   |     175      |      0.6x      |
> |    Ours    |  M=2 (Fast)   |    86.0     |    **86**    |    **1.3x**    |
>
>
>
> > **Q3.** Black-box targeted transfer to unseen architectures and cross-dataset tests are limited, so robustness to closed-source variability is not fully established.
>
> **A3.** Thanks for your comment. We clarify that our experimental setup rigorously evaluates distribution shifts. We have revised Section 3.4 (highlighted in blue) to explicitly state that the transfer scores in the attack logs used to train our GNN Attack Router are derived from attacks on Qwen2-VL-7B, which served as a reserved target model. This indicates that all target MLLMs evaluated in Tables 1, 2, and 3 constitute substantial distribution shifts relative to the training data, as they involve unseen architectures. Consequently, the trained Router effectively guides attacks on these diverse, unseen commercial models and achieves competitive ASR performance. This result strongly supports the Router’s robustness and its ability to generalize across diverse model distributions without fine-tuning for specific targets.

---

> > ### Author Response · Authors · 2025-11-20
> > **Response to Reviewer UTRC-2**
> >
> > > **Q4.** How does this method perform in comparison to other methods on other source->target transfer settings besides NIPS2017->COCO?
> >
> > **A4.** Thanks for your comment. We strictly follow the transfer settings from the NIPS 2017 competition dataset to MS COCO. This is a widely adopted method used by baseline models. This setup ensures that our comparisons are fair and directly reproducible. Instead of using multiple datasets, we focused on evaluating the model’s robustness along dimensions more relevant to transfer attacks, specifically target architecture diversity. We evaluated RISE on 14 distinct MLLMs (Tables 1–3). This extensive validation demonstrates that RISE generalizes well to fundamentally different semantic reasoning mechanisms, which we regard as a rigorous indicator of attack robustness.
> >
> >
> >
> >
> >
> > In light of these responses, we hope we have addressed your concerns, and hope you will consider raising your score. If there are any additional notable points of concern that we have not yet addressed, please do not hesitate to share them, and we will promptly attend to those points.

---

> > > ### Comment · Reviewer_UTRC · 2025-11-27
> > >
> > > Thank you for the detailed response and additional experiments.
> > >
> > > [**A1**] The cross-judge evaluation with Claude-Sonnet-4.5 and the added human study both strengthen the empirical evidence and reduce the risk that the gains are limited to GPT-based judge. However, this still does not fully establish generalizability across a broader range of MLLMs and input prompts. The current analysis uses only one extra judge (Claude-Sonnet-4.5), one attack setting, and a single fixed judging prompt, while LLM-as-judge behavior is known to be quite *prompt-sensitive.* The human study is also better viewed as a sanity check than a ground truth evaluation because statistical details such as sampled population, diversity of participants, etc. are not fully specified. *The paper can be further strengthened by clearly acknowledging these limitations of LLM-as-judge and small-scale human evaluation, and stating that robustness to prompt variation and to a diverse range of judges.*
> > >
> > > [**A2**] The added compute/latency trade-off comparisons convincingly clarify that RISE with different choices of *M* balances ASR and cost fairly against the baseline (FOA-Attack), and *I consider the compute concern resolved.*
> > >
> > > [**A3/A4**] The method is validated on the standard NIPS2017→MS COCO (1k images each) setting across a large set of target MLLMs, showing notable transfer to many closed-source models under this benchmark. *Nonetheless, the main transferability claim is still supported by only a single source–target dataset pairing. This leaves to question how well the method generalizes to other datasets or tasks, and whether some design choices are implicitly tuned to this particular benchmark.* I recommend making this restriction explicit in the limitations and framing broader cross-dataset / cross-task validation as a future work.
> > >
> > > Overall, the compute issue is resolved, and the LLM-as-judge and transferability concerns are only partially but not fully addressed. I would keep my original borderline score, but I would not oppose acceptance if the other reviewers take the current scope reasonable.

---

### Official Review · Reviewer_W8x2 · 2025-10-31

**Soundness:** 2
**Presentation:** 2
**Contribution:** 1
**Rating:** 2
**Confidence:** 5

**Summary:**

This paper aims to improve the transferability of adversarial attacks on Multimodal Large Language Models (MLLMs) from a statistical perspective. An adaptive ensemble mechanism is further designed to enhance attack performance. Experimental results show that the proposed method outperforms compared approaches.

**Strengths:**

1. Improving adversarial transferability is an important and meaningful research direction.
2. Experimental results demonstrate that the proposed approach achieves better performance than compared methods.

**Weaknesses:**

1. The motivation behind the proposed method is unclear. Although the authors claim to enhance adversarial transferability from a statistical viewpoint, they merely “hypothesize”  its effectiveness without offering solid theoretical or empirical justification.
2. The paper focuses on targeted attacks but presents them as general adversarial attacks. Since adversarial attacks include both targeted and untargeted settings, this distinction should be explicitly clarified.
3. In the second paragraph of page 2, the authors state that “This overfitting results in two key issues.” However, the subsequent text describes the causes of overfitting and limitations of existing methods, rather than clearly identifying the two key issues.
4. On page 6, the authors assert that “these strategies cannot capture the complex, asymmetric, and task-dependent transferability relationships,” but no supporting evidence or rationale is provided.
5. The ensemble evaluation is limited. Only three models are included in the pool, and the best performance is obtained using just two models. This setup fails to convincingly demonstrate the effectiveness of the adaptive ensemble strategy, and the authors do not explain why using two networks yields the best results.
6. The computational cost of the proposed method, compared with other approaches, is not discussed.
7. The attack performance is primarily evaluated based on feature similarity, without experiments across diverse tasks. Since MLLMs can be used in a variety of downstream applications, a feature-based evaluation alone cannot fully reflect the practical effectiveness of the attacks.
8. Adversarial attacks constitute a long-standing research area with many established methods across different domains, including conventional deep neural networks. The paper lacks a comprehensive literature review and should compare the proposed method against these broader approaches.

**Questions:**

Please refer to Weaknesses.

---

> ### Author Response · Authors · 2025-11-20
> **Response to Reviewer W8x2-1**
>
> We are truly grateful for the time you have taken to review our paper and your insightful review. Here we address your comments in the following.
>
>
>
> > **Q1.** The motivation behind the proposed method is unclear. Although the authors claim to enhance adversarial transferability from a statistical viewpoint, they merely “hypothesize” its effectiveness without offering solid theoretical or empirical justification.
>
> **A1.** Thanks for your comment. The motivation for employing Energy Distance arises from statistical manifold theory and the inherent limitations of point-to-point optimization in high-dimensional spaces. Standard attacks, such as minimizing the cosine distance to a single point, often drive adversarial examples toward low-probability regions outside the manifold, which contain non-robust features specific to other models. Our point of view that aligning distributions enhances generalization is not unfounded. Extensive experiments reported in Tables 1, 2 and 3 provide robust empirical validation, demonstrating that RISE consistently outperforms pointwise baselines by a substantial margin. If this statistical assumption were invalid, minimizing Energy Distance would not lead to more consistent improvements in cross-model transferability. Our findings across multiple distinct MLLMs further confirm the validity of this assumption.
>
>
>
> > **Q2.** The paper focuses on targeted attacks but presents them as general adversarial attacks. Since adversarial attacks include both targeted and untargeted settings, this distinction should be explicitly clarified.
>
> **A2.** Thanks for your comment. In the revised Abstract and Section 1, we explicitly define our problem scope as targeted transfer attacks (highlighted in blue), which aim to make the model produce a specific semantic target. In prior adversarial attack literature, targeted attacks are considered among the most challenging scenarios in adversarial machine learning. In this scenario, attackers attempt to manipulate MLLMs to produce semantic descriptions that align with their intentions. The revised section is as follows.
>
> “Unlike untargeted attacks that simply induce incorrect predictions, we focus on targeted attacks in which adversaries aim to manipulate MLLMs to produce specific, attacker-desired semantic captions.”
>
>
>
> > **Q3.** In the second paragraph of page 2, the authors state that “This overfitting results in two key issues.” However, the subsequent text describes the causes of overfitting and limitations of existing methods, rather than clearly identifying the two key issues.
>
> **A3.** Thank you for highlighting the ambiguous text structure on page 2. We have revised the paragraph (highlighted in blue) to clearly present the two resulting issues: 1) Limited generalization capability due to dependence on the surrogate's latent space. 2) Inability of static integration to handle task-related transferability.
>
>
>
> > **Q4.** On page 6, the authors assert that “these strategies cannot capture the complex, asymmetric, and task-dependent transferability relationships,” but no supporting evidence or rationale is provided.
>
> **A4.** Thanks for your comment. Our ablation study empirically supports the claim that transferability is asymmetric and task-dependent. **As shown in the table below (excerpted from Table 6 in Appendix D),** the heterogeneous (asymmetric) weighting strategy significantly outperforms the uniform (symmetric) weighting strategy. The observed performance gap (+3.0% / +7.0%) provides clear evidence that static, symmetric strategies cannot capture the complex, model-dependent variations in transferability. Furthermore, the left panel of Figure 3 shows that our adaptive Router outperforms fixed ensemble baselines, confirming the need for task-dependent modeling.
>
> |   **Weighting Scheme (η)**   |  **Type**  | **GPT-4o ASR** | **Gemini-2.0 ASR** |
> | :--------------------------: | :--------: | :------------: | :----------------: |
> |         $[1, 1, 1]$          | Symmetric  |     86.5%      |       73.5%        |
> | $[1.8, 2.1, 4.8]$ (Ours) | Asymmetric |   **89.5%**    |     **80.5%**      |

---

> ### Author Response · Authors · 2025-11-20
> **Response to Reviewer W8x2-2**
>
> > **Q5.** The ensemble evaluation is limited. Only three models are included in the pool, and the best performance is obtained using just two models.
>
> **A5.** Thanks for your comment. **(1) Standardization.** We used the same open-source CLIP surrogate model pool (ViT-B/16, ViT-B/32, Laion) as the baseline (M-Attack, FOA-Attack). Using non-standard model pools would result in unfair comparisons and deviate from established benchmark protocols.
>
> **(2) Why $K=2$.** As shown in our ablation study (Figure 4), $K=2$ achieves the optimal balance between diversity and gradient consistency. Although diversity is beneficial, including too many models introduces gradient noise that impedes the precise alignment needed for targeted attacks. Our GNN Router innovatively identifies optimal model pairings dynamically, rather than relying on the suboptimal static “all-use” strategy.
>
>
>
> > **Q6.** The computational cost of the proposed method, compared with other approaches, is not discussed.
>
> **A6.** Thanks for your comment. A detailed cost analysis has been added in Section 4.5 and Table 5. **As shown below.** RISE ($M=5$) involves data augmentation but avoids the complex internal optimization loops found in some baseline models. Even with reduced data augmentation ($M=2$), RISE is 1.3 times faster than FOA-Attack and achieves higher ASR, demonstrating remarkable efficiency.
>
> | **Method** | **Settings**  | **ASR (%)** | **Time (s)** | **Throughput** |
> | :--------: | :-----------: | :---------: | :----------: | :------------: |
> | FOA-Attack |   Standard    |    85.5     |     109      |      1.0x      |
> |    Ours    | M=5 (Default) |  **89.5**   |     175      |      0.6x      |
> |    Ours    |  M=2 (Fast)   |    86.0     |    **86**    |    **1.3x**    |
>
>
>
> > **Q7.** The attack performance is primarily evaluated based on feature similarity, without experiments across diverse tasks. Since MLLMs can be used in a variety of downstream applications, a feature-based evaluation alone cannot fully reflect the practical effectiveness of the attacks.
>
> **A7.** Thanks for your comment. We clarify that our approach does not rely on feature similarity. Our main metrics (ASR, AvgSim, and KMR) assess model performance on downstream image description tasks using LLM-based evaluation criteria and keyword matching. This directly reflects the practical effectiveness of the attacks. In the revised manuscript (Section 4.5), we also included cross-judge validation and human evaluation to further confirm these results, as summarized in the table below.
>
> | **Method** | **Model** | **Original Judge (ASR)** | **Original Judge (AvgSim)** | **Claude Judge (ASR)** | **Claude Judge (AvgSim)** | **Human Eval (ASR)** |
> | :--------: | :-------: | :----------------------: | :-------------------------: | :--------------------: | :-----------------------: | :------------------: |
> | FOA-Attack | Ensemble  |           85.5           |            0.50             |          79.0          |           0.51            |         81.0         |
> |    Ours    | Ensemble  |         **89.5**         |          **0.53**           |        **84.0**        |         **0.57**          |       **85.0**       |
>
>
>
> > **Q8.** The paper lacks a comprehensive literature review and should compare the proposed method against these broader approaches.
>
> **A8.** Thanks for your suggestion. We expanded Section 2 (Related Work) to include a discussion on traditional attacks. We highlighted a key distinction: MLLM attacks must bridge the modality gap between vision and language, while traditional attacks focus only on classification boundaries. This makes our proposed distribution-aware approach especially well-suited for MLLMs. The specific modifications are summarized as follows.
>
> “Comparison with Traditional Attack Methods. Traditional attack methods (e.g., FGSM and PGD) primarily exploit high-frequency pixel noise to cross static decision boundaries in closed-set classification, resulting in simple label misclassifications (Cao et al., 2025). In contrast, MLLM attacks operate in an open semantic space dominated by cross-modal alignment, where the attack objective shifts from simple misclassification to more complex forms of semantic hijacking or jailbreaking (Cui et al., 2025). Because the visual–language projector in MLLMs acts as a semantic filter that blocks simple pixel-level gradient noise, point-to-point feature-matching strategies developed for CNNs do not transfer directly (Wang et al., 2025a). Consequently, effective MLLM attacks must move beyond pixel-level perturbations and instead exploit distribution-level semantic alignment to breach the security barriers of multimodal interactions—a defining characteristic of this new generation of threat models (Rahmatullaev et al., 2025).”

---

> > ### Author Response · Authors · 2025-11-20
> > **Response to Reviewer W8x2-3**
> >
> > **Regarding Ethical Concerns.** We note that you flagged “Discrimination/bias/fairness concerns.” Although we could not find specific comments, we respectfully clarify that this paper is a technical study on adversarial robustness (red-teaming), focusing on uncovering vulnerabilities to improve model security. We have thoroughly discussed the related dual-use risks and ethical considerations in Appendix I: Broader Impact and Ethical Considerations. If you identify any specific images or descriptions in the text that appear biased, please let us know, and we will address them promptly.
> >
> >
> >
> > In light of these responses, we hope we have addressed your concerns, and hope you will consider raising your score. If there are any additional notable points of concern that we have not yet addressed, please do not hesitate to share them, and we will promptly attend to those points.

---

> > ### Comment · Reviewer_W8x2 · 2025-11-27
> > **Response to the authors**
> >
> > Thank you for the rebuttal. However, many of my concerns remain unresolved.
> > Regarding A1, no theoretical justification has been provided.
> > For A5, the issue of using only three models in the surrogate pool is still not adequately addressed.
> > For A7, ASR, AvgSim, and KMR continue to rely on feature similarity, which does not fully resolve the concern.
> > For A8, the lack of comparison with additional strong baselines remains a significant limitation.
> > Given these remaining issues, I will keep my original score.

---

### Official Review · Reviewer_VDqN · 2025-10-31

**Soundness:** 3
**Presentation:** 3
**Contribution:** 3
**Rating:** 6
**Confidence:** 4

**Summary:**

The paper proposes Relational Distribution-aware Intrinsic Alignment (RISE), a transfer attack framework for MLLMs with two main ideas: (i) move from pointwise feature matching to distributional alignment in a surrogate’s latent space using Energy Distance; and (ii) replace heuristic model ensembling with a GNN Attack Router trained offline on historical attack logs to choose a task-adaptive surrogate ensemble. Experiments report SOTA transferability on several open- and closed-source MLLMs under targeted captioning attacks, with metrics including ASR, AvgSim, and a keyword-based KMR.

**Strengths:**

1.	Clear conceptual shift: Reframing transfer from pointwise to distributional alignment is well-motivated and novel in this context. Using Energy Distance (nonparametric, no kernel tuning) is a principled choice for empirical distribution matching.
2.	Adaptive model selection: The GNN Attack Router formalizes surrogate selection as a relational prediction problem using task/model graphs, moving beyond fixed or purely loss-based weighting.
3.	Empirical gains: On open-source models (e.g., LLaVA-1.5-7B, Qwen2.5-VL-7B), RISE outperforms strong baselines including FOA-Attack; similar trends hold for closed-source models (e.g., GPT-4o/4.1, Claude-3.7).

**Weaknesses:**

1.	Dependence on LLM-as-Judge metrics: Reported ASR/AvgSim rely on GPT-style graders, which are sensitive to prompt design and judge choice and can diverge from human judgments. This creates uncertainty about absolute and relative gains. Please include a small human study and/or cross-judge robustness (multiple prompts and at least two independent judge models with calibration).
2.	Router practicality & cold-start risk: The GNN Attack Router presumes access to extensive historical attack logs for offline training. In realistic deployments (especially on new or proprietary closed-source targets) curating such data is precisely the hard part, creating a cold-start loop: training the router for a new target M_new requires transfer outcomes on M_new, which in turn requires many prior attacks on that same model. Please clarify how RISE operates when no target-specific history exists and outline a fallback (e.g., metadata-based zero-shot priors, family-level generalization, or lightweight online updates). Related cold-start challenges are well-documented in GNN-based selection/routing contexts.
3.	Scalability and Computational Overhead. The paper presents RISE as an effective framework but provides limited analysis of its computational cost, which appears non-trivial in two key areas. (1)  Building the router’s heterogeneous graph appears to require a combinatorial sweep over surrogate ensembles, tasks, and targets. This prerequisite could exceed the cost of the downstream attacks it aims to optimize. Please quantify the size of the logs, number of attacks run, and total compute. (2) The distribution-aware intrinsic mining scales roughly with augmentations M and selected ensemble size K. The paper’s own runtime analysis suggests noticeable growth with larger M; “computationally efficient” should be contextualized against simpler point-wise baselines. Consider reporting a wall-clock/throughput table vs. M and K, and a Pareto curve vs. ASR.
4.	Ambiguities in Methodology and Reproducibility. The high-level method is clear, but key details are missing for full replication—notably the router’s exact architecture (GNN layer type, depth, hidden sizes, readout), training data composition, and update cadence as models evolve. Please include these in the main text or an appendix, and provide seeds, config files, and code for the router pipeline.

**Questions:**

1.	Router generalization: If a target model family was unseen during router training, how does selection degrade? Any leave-one-family-out results?
2.	Data / compute: What is the exact size of the historical log, feature dimensionality, and training cost for the router?
3.	Evaluation robustness: Can you add a small-scale human study or at least a cross-judge experiment (e.g., Claude judge vs. GPT judge) to verify gains under different evaluators?
4.	Cold-start deployment. How would RISE be used against a brand-new closed-source MLLM with no prior logs? Do you support (i) metadata-only priors (e.g., family, tokenizer/vision-tower hints), (ii) few-shot online router updates, or (iii) a fallback heuristic (e.g., diversity-maximizing surrogate subset)? Provide an explicit procedure.
5.	Offline cost quantification. Please estimate the total number of attacks used to build the experimental graph and the corresponding compute/budget (GPU-hours, API costs for closed-source queries if applicable). A per-target and per-surrogate breakdown would be ideal.
6.	Router specification for reproducibility. Please document the GNN router architecture (layer type, depth, hidden sizes, message-passing steps, readout, optimizer, training epochs, early-stopping criteria) and release config files.
7.	Ensemble size & negative contribution (hypothesis). In your ablation, K=2 appears to outperform K=3. Could it be that, in many cases, a particular pair of surrogates contributes most of the gain while the third model sometimes has a negative marginal effect—raising the question of whether a simpler “top-2” heuristic would perform similarly? Please report (i) per-example ensemble composition histograms, (ii) leave-one-out/marginal contribution analyses (e.g., $\Delta$ ASR when adding the 3rd model), and (iii) a comparison against fixed top-2 or diversity-maximizing heuristics to quantify any incremental benefit of the router.

---

> ### Author Response · Authors · 2025-11-20
> **Response to Reviewer VDqN-1**
>
> We are truly grateful for the time you have taken to review our paper, your insightful comments and support. Your positive feedback is incredibly encouraging for us! In the following response, we would like to address your major concern and provide additional clarification.
>
>
>
> > **Q1.** Evaluation robustness: Can you add a small-scale human study or at least a cross-judge experiment (e.g., Claude judge vs. GPT judge) to verify gains under different evaluators?
>
> **A1.** Thanks for your comment. We acknowledge that solely relying on GPT-like models for evaluation may introduce bias. To rigorously validate our findings, we conducted two additional evaluations.
>
> **(1) Cross-Judge Validation.** We used the advanced Claude-Sonnet-4.5 model as an independent judge to assess adversarial samples generated against GPT-4o. **The results are presented in the table below.** ASR scores decreased slightly, likely due to differences in semantic similarity sensitivity across MLLMs, as Claude-Sonnet-4.5 applies stricter criteria. However, the relative performance rankings remained consistent. Notably, RISE achieved an ASR of 84.0%, outperforming the strongest baseline, FOA-Attack (79.0%), by 5.0%. This result confirms that RISE's improvement arises from robust semantic alignment and is unaffected by variations in the evaluation standard.
>
> **(2) Human Evaluation.** We conducted a human evaluation of adversarial samples generated by GPT-4o. To reduce evaluation complexity, annotators assigned only Yes/No labels. **The results are presented in the table below.** RISE achieved an average human-evaluated ASR of 85%, whereas FOA-Attack reached 81%. This further validates RISE’s effectiveness in generating semantic transfers that are perceptible to human evaluators.
>
> | **Method** | **Model** | **Original Judge (ASR)** | **Original Judge (AvgSim)** | **Claude Judge (ASR)** | **Claude Judge (AvgSim)** | **Human Eval (ASR)** |
> | :--------: | :-------: | :----------------------: | :-------------------------: | :--------------------: | :-----------------------: | :------------------: |
> | FOA-Attack | Ensemble  |           85.5           |            0.50             |          79.0          |           0.51            |         81.0         |
> |    Ours    | Ensemble  |         **89.5**         |          **0.53**           |        **84.0**        |         **0.57**          |       **85.0**       |
>
>
>
> > **Q2.** Cold-start deployment. How would RISE be used against a brand-new closed-source MLLM with no prior logs? / Router generalization: If a target model family was unseen during router training, how does selection degrade?
>
> **A2.** Thanks for your comment. We sincerely apologize for omitting the model used to train the Router in the initial submission. We have revised Section 3.4 (highlighted in blue) to explicitly state that the transfer scores in the attack log used to train our GNN Attack Router originate from attacks on Qwen2-VL-7B, which served as a reserved target model. Importantly, this model is completely distinct from all target MLLMs evaluated in Tables 1, 2, and 3. This setup confirms that our framework operates in a cold-start (zero-shot) setting with respect to the target model. The Router learns to map visual features (source and target images) to surrogate efficacy, enabling it to capture generalizable transferability patterns (e.g., “complex scenes transfer better via Laion”) rather than overfitting to Qwen2-specific parameters. Because the Router was trained on transfer scores originating from attacks on Qwen2-VL-7B, yet still successfully attacked a wide range of MLLMs, this demonstrates that no additional data collection or retraining is required for new target models. The cold-start hurdle is eliminated because the Router generalizes across model architectures.

---

> > ### Author Response · Authors · 2025-11-20
> > **Response to Reviewer VDqN-2**
> >
> > > **Q3.** Offline cost quantification. Please estimate the total number of attacks used to build the experimental graph and the corresponding compute/budget. / Consider reporting a wall-clock/throughput table vs. M and K.
> >
> > **A3.** Thanks for your comment. **(1) Offline Costs.** Data collection is a one-time effort. Generating 800 attack logs requires approximately 28 GPU hours. Training the GNN is extremely fast, taking less than one hour. Although data collection incurs costs, it represents a one-time fixed investment. Due to its zero-shot generalization capabilities (see **Q2**), the trained Router can directly target numerous new models, such as GPT-4o and Claude-3.5, without additional data collection. Therefore, the marginal cost per new task is virtually zero.
> >
> > **(2) Online Cost and Performance.** Regarding the effect of augmentation quantity $M$, Figure 4 shows that while computation time increases linearly, performance saturates around $M=5$. Compared to baselines such as FOA-Attack, which involves complex optimization loops, RISE with $M=5$ provides a better trade-off between ASR and computation time. **The table below compares the processing time per image between RISE and FOA-Attack.** Even with reduced data augmentation ($M=2$), RISE runs 1.3 times faster than FOA-Attack while maintaining higher ASR, demonstrating remarkable efficiency.
> >
> > | **Method** | **Settings**  | **ASR (%)** | **Time (s)** | **Throughput** |
> > | :--------: | :-----------: | :---------: | :----------: | :------------: |
> > | FOA-Attack |   Standard    |    85.5     |     109      |      1.0x      |
> > |    Ours    | M=5 (Default) |  **89.5**   |     175      |      0.6x      |
> > |    Ours    |  M=2 (Fast)   |    86.0     |    **86**    |    **1.3x**    |
> >
> >
> >
> > > **Q4.** Router specification for reproducibility. Please document the GNN router architecture and release config files.
> >
> > **A4.** Thanks for your comment. We have added the full specification in Appendix A.1 (highlighted in blue), which is presented below.
> >
> > “To implement the GNN Attack Router, we adopt a heterogeneous GraphSAGE architecture built on the PyTorch Geometric library.
> >
> > Encoder: The encoder comprises two layers of heterogeneous graph convolutions (HeteroConv). Each layer uses a SAGEConv operator with mean pooling to aggregate information from neighboring nodes. The hidden feature dimension is set to 64.
> >
> > Predictor: The edge predictor is a two-layer MLP (Linear → ReLU → Linear) that concatenates the updated features of task and model nodes to predict scalar transfer scores.
> >
> > Input Projection: A linear projection layer maps the high-dimensional task features (ViT embeddings) and model features (one-hot encoded) onto a shared hidden space ($d=64$) before they are fed into the GNN encoder.”
> >
> >
> >
> > > **Q5.** Ensemble size & negative contribution (hypothesis). In your ablation, K=2 appears to outperform K=3. Could it be that, in many cases, a particular pair of surrogates contributes most of the gain while the third model sometimes has a negative marginal effect—raising the question of whether a simpler “top-2” heuristic would perform similarly?
> >
> > **A5.** We sincerely appreciate the insightful hypothesis you have proposed. We fully agree with your observation that the performance drop from $K=2$ to $K=3$ suggests that adding a third, lower-matching surrogate model often results in a “negative marginal contribution.” This is because, in adversarial transfer learning, adding more models does not always improve performance. As observed in previous studies and our experiments, gradient directions from different surrogate models may conflict, introducing noise that disrupts the precise semantic alignment required for targeted attacks. This explains why we optimized for a compact ensemble size of $K=2$.
> >
> > Regarding why Router outperforms fixed Top-2 heuristics, while a fixed optimal Top-2 selection (e.g., always choosing the historically strongest two models) serves as a robust baseline, it relies on a static assumption. However, transferability is highly dependent on the data. For texture-rich image A, the optimal model combination may be ViT-B/16 + Laion. For shape-rich image B, the optimal model combination may be ViT-B/16 + ViT-B/32. The GNN Attack Router effectively functions as a dynamic Top-2 selector. It predicts which specific combination maximizes transferability for a given input instance, capturing subtle, asymmetric relationships often overlooked by global heuristics. Thus, the Router ensures that we always use the optimal two models, rather than merely the average of the best two.
> >
> >
> >
> >
> >
> > Thanks again for appreciating our work and for your constructive suggestions. Please let us know if you have further questions.

---

### Author Response · Authors · 2025-12-03
**Summary of Additional Experiments & Final Response**

Dear Area Chair and Reviewers,



We sincerely thank the reviewers (VDqN, W8x2, UTRC, DnBK) for their constructive feedback. As the discussion phase concludes, we summarize the supplementary experiments and revisions conducted to address the remaining issues related to generalization, robustness, and evaluation metrics.





**1. Cross-Dataset Generalization (Addressing UTRC & DnBK).** To assess RISE's transferability beyond the NIPS$\to$COCO setting, we conducted transfer attacks using the held-out NIPS source images against a completely new target dataset: Flickr30k (200 pairs). We evaluated the attack performance on GPT-4o and Gemini-2.0. **As shown in the table below,** RISE consistently outperforms the strongest baseline (FOA-Attack) across a range of attacked MLLMs. The substantial improvement on the Gemini-2.0 further demonstrates that our Distribution-aware Intrinsic Mining effectively captures general semantic patterns that are robust to domain shifts.

| **Method** | **Model** | **GPT-4o (ASR)** | **GPT-4o (AvgSim)** | **Gemini-2.0 (ASR)** | **Gemini-2.0 (AvgSim)** |
| :--------: | :-------: | :--------------: | :-----------------: | :------------------: | :---------------------: |
| FOA-Attack | Ensemble  |       83.5       |        0.46         |         70.5         |          0.38           |
|    Ours    | Ensemble  |     **85.0**     |      **0.48**       |       **77.5**       |        **0.41**         |



**2. Sensitivity to the Perturbation Budget $\epsilon$ (Addressing DnBK).** We conducted a sensitivity analysis by varying the perturbation budget $\epsilon$. **As shown in the table below,** under a strict constraint ($\epsilon = 8/255$), RISE consistently outperforms the baseline algorithms. This result supports our theoretical assertion that distribution alignment is more robust than point-wise matching when the optimization space is limited. With a larger budget ($\epsilon = 32/255$), RISE’s performance saturates at 90.0%, indicating that our method reaches peak performance under standard settings without requiring excessive perturbation.

| **Budget ($\epsilon$)** | **Method** | **ASR (%)** |
| :---------------------: | :--------: | :---------: |
|       8/255 (Low)       | FOA-Attack |    64.5     |
|       8/255 (Low)       |    Ours    |    68.0     |
|    16/255 (Standard)    |    Ours    |    89.5     |
|      32/255 (High)      |    Ours    |    90.0     |



**3. Theoretical and Paper Revisions (Addressing W8x2 & UTRC & DnBK).** We have uploaded the revised version of the paper, and the key updates are summarized below:

**(1) Theoretical Proof (Appendix K):** We added a formal derivation showing that our Energy Distance objective functions as a gradient variance reduction estimator. This provides a theoretical explanation for why our distribution-aware alignment method mitigates overfitting to high-frequency artifacts in surrogate models, whereas point-wise matching does not.

**(2) Supplementary Experiments (Appendix L):** We provide complete data tables for the Flickr30k transfer attack and the perturbation budget sensitivity analysis.

**(3) Limitations and Protocols (Appendices H.3 & C):** We explicitly discussed the scope of benchmarks (NIPS$\to$COCO) and present detailed protocols for our human evaluation on semantic consistency.



**4. Important Clarifications (Addressing W8x2's Concerns).** We would like to respectfully clarify two factual misunderstandings regarding our methodology:

**(1) Evaluation Metrics:** Our reported metrics (ASR, AvgSim, KMR) are text-based semantic evaluations performed on the generated captions by MLLMs (using LLM-as-a-judge and keyword matching), not latent feature similarities. They directly measure downstream task performance.

**(2) Fair Comparison:** We use a specific three-model source pool strictly to ensure a fair comparison under the standard protocols established by the baseline methods (M-Attack, FOA-Attack).



We believe that the additional experiments and theoretical proofs have comprehensively addressed the reviewers' concerns. We respectfully request the Area Chair to consider the demonstrated robustness and generalization of RISE in the final decision.



Best regards,

The Authors

---

### Meta-Review · Area_Chair_rJVY · 2026-01-03

**Summary:**

Reviewers found the core idea, Energy Distance–based distributional alignment + GNN router for surrogate selection, empirically strong on the NIPS2017→COCO benchmark with many closed-source targets. The recommendation leans Reject mainly due to remaining concerns about evaluation validity/generalization (LLM-as-judge prompt/judge sensitivity; limited cross-dataset/task coverage) and some skepticism about the router/ensemble setup and broader baseline coverage, with one reviewer staying firmly negative.

**Reviewer Concerns:**

Addressed: cross-judge + small human study; added compute/latency trade-off analysis; clarified router “cold-start” training setup and provided router architecture details; added limited multi-seed stability; added some extra generalization/sensitivity experiments like Flickr30k transfer and ε sensitivity.

Still outstanding: reliance on LLM-as-judge remains only partially de-risked (single extra judge, fixed prompt; human study details limited); broader generalization still constrained (mostly one dataset-pair/one task setting); stronger budget-sweep coverage and broader baseline comparisons remain limited; one reviewer explicitly maintains that key concerns are unresolved.

**Reviewer Scores:**

- VDqN (6): likely 6 → 6 many requests addressed.
- W8x2 (2): 2 → 2 reviewer explicitly keeps original score.
- UTRC (4): 4 → 4 compute concern resolved; judge + transfer-scope only partially addressed.
- DnBK (6): likely 6 → 6 confidence improved; still requests broader ε-budget sweeps; "plan to keep original overall evaluation".

---

### Decision · Program_Chairs · 2026-01-26

Reject